# OpenReview forum: "ODESteer: A Unified ODE-Based Steering Framework for LLM Alignment"
_ICLR.cc/2026/Conference — ICLR 2026 Poster_

### Official Review · Reviewer_s7Ss · 2025-10-29

**Soundness:** 3
**Presentation:** 3
**Contribution:** 3
**Rating:** 6
**Confidence:** 3

**Summary:**

This paper provides a unified theoretical framework based on Ordinary Differential Equations (ODEs) for activation steering.

The authors show that conventional one-step activation addition, $\tilde{a} = a + T \cdot v(a)$, can be rigorously interpreted as the first-order Euler discretization of an ODE, $\dot{a}(t)=v(a(t))$. Within this framework, they argue that identifying the optimal steering direction $v(a)$ is equivalent to designing a barrier function, $h(a)$, which separates desirable and undesirable regions in the activation space, guiding the activations toward desirable regions while ensuring it remains there.

Building on this, the authors introduce BODES. BODES defines the barrier function $h(a)$ as the log-density ratio between positive and negative activations, utilizing nonlinear features (specifically, Polynomial Count Sketch) to capture complex patterns. By solving the ODE derived from the normalized gradient of this nonlinear barrier function, BODES performs multi-step, adaptive steering where the direction changes dynamically based on the current activation. Empirically, BODES achieves consistent improvements: 7% on TruthfulQA, 2% on RealToxicityPrompts, and 2% on UltraFeedback.

**Strengths:**

1. The theoretical framework is novel. Interpreting activation steering (a discrete intervention) as the numerical solution (Euler discretization) of a continuous Ordinary Differential Equation (ODE) is new and overcome the limitation of previous methods in one-step mapping.

2. Empirical results are quite strong and significant. BODES achieves consistent improvements: 7% on TruthfulQA, 2% on RealToxicityPrompts, and 2% on UltraFeedback. Extensive experiments cover various LLMs and tasks.

3. The paper is clearly written.

**Weaknesses:**

1. Activation steering is valued for being "lightweight" and inference-time usable. BODES involves: (i) calculating nonlinear features (Polynomial Count Sketch), (ii) computing the Jacobian of this map, and (iii) running a 10-step numerical ODE solver at every generated token. While the components are individually efficient, the cumulative overhead might be substantial compared to a single vector addition (CAA) or matrix multiplication (SEA). Practically, it is beneficial to include a comparison of the average generation latency (e.g., tokens/second or wall-clock time per token) for BODES against the baselines (e.g., CAA or ITI compared in your paper).

CAA - Steering Llama 2 via Contrastive Activation Addition. ACL 2024.
SEA - Spectral editing of activation for LLM alignments. Neurips 2024.
ITI - Inference-Time Intervention: Eliciting Truthful Answers from a Language Model. Neurips 2023

2. The paper attributes performance gains to "improved numerical accuracy" from multi-step ODE solving (Section 5.3) and states the Euler method is used for 10 steps (Appendix C.3). However, in addition to the ablation for single-step BODES vs BODES, it would strengthen the paper if you could ablate on the performacne vs number of steps. And, trying a higher-order numerical methods may further verify the performance gain sourced from the numeircal accuracy.

3. BODES has a strong theoratical solidness but it inevitably introduce addition complexit in inference-time compute and hyperparemeter search, e.g., strength hyperparams listed in Table 5 seems to be sensitive with model choice.

**Questions:**

1. Table 5 gives the range of $T$ used for different models, given that $T$ controls the overall steering strength and is a critical hyperparameter, could you provide a plot (similar to Figure 2 for layer selection) showing the metric of interest (e.g., Win (%) for UltraFeedback) as a function of $T$ for one model/task pair (e.g., LLaMA3.1-8B on UltraFeedback)? This would illustrate the sensitivity of BODES to the key strength hyperparameter $T$.

2. The optimal intervention layer is selected based on the peak performance of a linear, one-step method (CAA, Fig. 2). Was this layer selection validated for the multi-step, and nonlinear BODES method? Given the difference in steering dynamics, is it guaranteed that the optimal layer for a simple linear steer (CAA) is also optimal for ODE-based steering?

---

> ### Author Response · Authors · 2025-11-21
> **Rebuttal by Authors**
>
> Dear Reviewer s7Ss,
>
> We sincerely thank you for taking the time to review our paper and for providing insightful and constructive feedback. We have carefully addressed all the points and suggestions you raised. Please feel free to let us know if you have any additional questions or concerns. We greatly appreciate the opportunity to further refine our work. Thank you again for your thoughtful review and valuable input.
>
> **Q 4.1: It is beneficial to include a comparison of the average generation latency (e.g., tokens/second or wall-clock time per token) for BODES against the baselines (e.g., CAA or ITI compared in your paper).**
>
> **A 4.1:** We compare the inference efficiency of BODES with other baseline methods on TruthfulQA. **The number of generated tokens per second** for each method is presented in **Tab 4.1**. The results show that the generation speed of BODES is **only slightly slower than the no-steering (original) condition**  and other one-step steering methods such as CAA, which is expected due to the multi-step nature of BODES.
>
> However, BODES is still **faster than several DNN-based steering methods**, including RE-Control and TruthFlow, which involve additional DNN computations during inference. These findings indicate that BODES maintains practical inference efficiency while introducing only minimal additional computational cost. We also include these results in Appendix E.2 (page 20, line 1069-1098) of the revised manuscript.
>
> **Tab 4.1 The number of generated tokens per second for three LLMs (higher is better).**
>
> | Model        | Falcon-7B     | Mistral-7B    | Llama3.1-8B   |
> |:-------------|:--------------|:--------------|:--------------|
> | Original     | 117.69 ± 0.45 | 116.26 ± 0.24 | 114.82 ± 0.18 |
> | RepE         | 117.69 ± 0.27 | 115.82 ± 0.08 | 114.71 ± 0.3  |
> | ITI          | 117.54 ± 0.12 | 115.78 ± 0.07 | 114.82 ± 0.29 |
> | CAA          | 117.46 ± 0.44 | 115.76 ± 0.03 | 114.57 ± 0.48 |
> | MiMiC        | 105.62 ± 0.36 | 109.66 ± 0.16 | 108.73 ± 0.38 |
> | HPR          | 116.09 ± 0.04 | 115.07 ± 0.12 | 114.42 ± 0.11 |
> | RE-Control   | 98.03 ± 0.51  | 101.05 ± 0.03 | 99.94 ± 0.11  |
> | LinAcT       | 117.61 ± 0.17 | 116.17 ± 0.03 | 115.0 ± 0.42  |
> | TruthFlow    | 62.45 ± 0.38  | 62.06 ± 0.46  | 62.33 ± 0.48  |
> | BODES (Ours) | 107.41 ± 0.22 | 105.89 ± 0.08 | 106.76 ± 0.06 |
>
> **Q 4.2: It would strengthen the paper if you could ablate on the performance vs number of steps. And, trying a higher-order numerical methods may further verify the performance gain sourced from the numerical accuracy.**
>
> **A 4.2:** We evaluate the effect of the number of integration steps on BODES performance using TruthfulQA. The results, shown in **Fig. 4 in Appendix E.4** of the revised manuscript, indicate that increasing the number of steps yields mild performance improvements at first, after which the performance becomes stable. This suggests that BODES reaches sufficient numerical accuracy with a relatively small number of steps. The limited additional gains are expected, as the learned barrier function already provides a reliable steering direction.
>
> We also assess the sensitivity of BODES to the choice of ODE solver. As reported in **Tab. 4.2**, BODES is almost insensitive to solver type: higher-order methods such as fourth-order Runge-Kutta (RK4) offer only marginal improvements over simpler solvers. Considering computational efficiency and conceptual simplicity, we therefore adopt the Euler method as our default solver.
>
> **Tab 4.2: The impact of ODE solver types on the performance of BODES on TruthfulQA.**
>
> | ODE Solver Type      | Falcon-7B-Base   | Mistral-7B-Base   | Llama3.1-8B-Base   |
> |:------|:-----------------|:------------------|:-------------------|
> | Euler | 42.2 ± 0.115     | 59.9 ± 0.237      | 63.2 ± 0.823       |
> | RK4   | 42.8 ± 0.555     | 60.2 ± 1.328      | 63.3 ± 0.923       |
>
> **Q 4.3: BODES inevitably introduces addition complexity in inference-time compute and hyperparemeter search, e.g., strength hyperparemeters listed in Table 5 seems to be sensitive with model choice.**
>
> **A 4.3:** First, we emphasize that BODES does not introduce significant computational overhead during inference, as shown in **Tab 4.1**.
>
> Second, although the intervention strength hyperparameter $T$ may vary across different LLMs, we argue that **it is the only hyperparameter that requires tuning in our method. All other components (including the settings of the polynomial count sketch and the ODE solver) are fixed across models and datasets**, as described in Appendices C.2 and C.3. Furthermore, once the intervention strength is determined for a given LLM (see Appendix C.3), it can generally be kept fixed across different datasets. **Compared with DNN-based steering methods such as RE-Control and TruthFlow, they require much more extensive hyperparameter tuning for each combination of LLM and dataset.** This demonstrates the practical usability of BODES.

---

> ### Author Response · Authors · 2025-11-21
> **Rebuttal by Authors (Continued)**
>
> **Q 4.4: Table 5 gives the range of T used for different models, given that T controls the overall steering strength and is a critical hyperparameter, could you provide a plot (similar to Figure 2 for layer selection) showing the metric of interest (e.g., Win (%) for UltraFeedback) as a function of T for one model/task pair (e.g., LLaMA3.1-8B on UltraFeedback)? This would illustrate the sensitivity of BODES to the key strength hyperparameter T.**
>
> **A 4.4:** To maintain consistency with our layer-selection analysis, we conducted a sensitivity study on the steering-strength parameter $T$ using Llama3.1-8B on TruthfulQA. The results are presented in **Fig. 5 in Appendix E.4** (page 22). We can observe from the figure that **BODES exhibits stable performance across the selected range of** $T$ (4-6). However, when T becomes excessively large, corresponding to overly strong interventions, performance degrades notably due to deterioration in generation quality. This behavior is consistent with observations from prior activation-steering methods such as ITI. In future work, we plan to explore approaches for automatically selecting an appropriate value of $T$ for BODES.
>
>
> **Q 4.5: The optimal intervention layer is selected based on the peak performance of a linear, one-step method (CAA, Fig. 2). Was this layer selection validated for the multi-step, and nonlinear BODES method? Given the difference in steering dynamics, is it guaranteed that the optimal layer for a simple linear steer (CAA) is also optimal for ODE-based steering?**
>
> **A 4.5:** We evaluated BODES across multiple intervention layers on the TruthfulQA dataset using Llama3.1-8B, as shown in **Fig. 6 in Appendix E.5** (page 23) of the revised manuscript. The results indicate that the optimal steering layer for BODES is slightly different from that of CAA (differing by only one layer). However,  we observe that the optimal layer still falls within the same region identified by CAA - namely, the middle of the model layers — which aligns with prior findings in activation steering.
>
> Importantly, our use of CAA for layer selection is intended to **ensure a fair and consistent comparison** across different steering methods. Notably, even when BODES is not applied at its individually optimized layer, it still **consistently outperforms SOTA steering baselines**.

---

> ### Author Response · Authors · 2025-11-25
> **Request for Follow-Up Feedback on Author Rebuttal**
>
> Dear Reviewer s7Ss,
>
> Our detailed rebuttal has been submitted, and we have thoroughly addressed all the points and suggestions you raised. We understand the significant workload involved in reviewing papers, but we kindly request your feedback on our responses to ensure that the discussions are as productive and comprehensive as possible. Your insights will be invaluable in refining the final version of our work. Thank you once again for your time and effort!

---

### Official Review · Reviewer_BqZv · 2025-11-01

**Soundness:** 2
**Presentation:** 3
**Contribution:** 3
**Rating:** 2
**Confidence:** 3

**Summary:**

The paper “Activation Steering for LLM Alignment via a Unified ODE-Based Framework (BODES)” proposes a principled way to steer large language model (LLM) activations toward desirable behaviors without retraining. The authors reinterpret prior *activation addition* techniques (which simply add a fixed direction vector to hidden states) as a special case of an Ordinary Differential Equation (ODE) system.
They define an **activation flow**:

$$\dot{a}(t) = \nabla_a \log \frac{p_+(a)}{p_-(a)}$$

where $(p_+(a)) and (p_-(a))$ are activation densities of *desired* and *undesired* behaviors.
Integrating this ODE (via an adaptive solver) moves activations smoothly toward regions of high $(p_+(a))$ — that is, *truthful, helpful, or harmless* behavior — while guaranteeing stability and forward invariance (once inside the “good” region, activations remain there from the proposition 1).

Experiments on multiple open LLMs show small but consistent gains (≈2–7%) in truthfulness and safety benchmarks,

**Strengths:**

1. The paper introduces a theoretically grounded and unified ODE-based framework for activation steering.
2. It enables adaptive, stable, and efficient alignment of LLMs without any retraining.
3. The method shows consistent improvements across truthfulness and safety benchmarks

**Weaknesses:**

1. **Minimal performance improvement:** The reported gains (≈2–7%) are small and may not be statistically significant.
2. **Lack of clarity on \(p_+\) and \(p_-\):** The paper doesn’t specify how positive and negative activations are categorized or sampled.
3. **Unverified barrier property:** It is not shown that the learned barrier function \(h(a)\) satisfies Proposition $\(\dot{h}(a)=\nabla h(a)^\top v(a)>0\)$ in practice.
4. **No trajectory-level likelihood analysis:** The paper doesn’t measure or visualize how average likelihood (\(h(a)\)) changes along the ODE trajectory.

**Questions:**

1. How are $p_+$ and $p_-$ activations defined ?
2. Does activation steering actually increase the *average likelihood* along trajectories compared to unsteered activations?
3. Can the authors verify that the barrier condition \(\dot{h}(a) > 0\) holds for the learned \(h(a)\)?
4. Could this framework generalize to  alignment to reward-based alignment , comparison with self consistency

---

> ### Author Response · Authors · 2025-11-21
> **Rebuttal by Authors**
>
> Dear Reviewer BqZv,
>
> We sincerely thank you for taking the time to review our paper and for providing insightful and constructive feedback. We have carefully addressed all the points and suggestions you raised. Please feel free to let us know if you have any additional questions or concerns. We greatly appreciate the opportunity to further refine our work. Thank you again for your thoughtful review and valuable input.
>
> **Q 3.1: Minimal performance improvement: The reported gains (≈2–7%) are small and may not be statistically significant.**
>
> **A 3.1:** First, **as clearly stated in line 123-125**, we reiterate that the primary contribution of our work lies in **establishing a unified theoretical framework for activation steering**. This framework provides principled insights that can guide future analysis and the systematic design of activation steering.
>
> Second, **the reported gains are already competitive within the activation-steering literature**. The baseline methods we compare against, such as MiMiC (ICML 2024), LinAcT (ICLR 2025 Spotlight), and TruthFlow (ICML 2025), typically achieve improvements of similar or smaller magnitude, and our method consistently outperforms these baselines across tasks and models. **We further note that other reviewers (cs8e and s7Ss) have explicitly commented that the improvements are meaningful in the context of activation steering.**
>
>
> **Q 3.2: Lack of clarity on ($p_+$) and ($p_-$): The paper doesn’t specify how positive and negative activations are categorized or sampled.**
>
> **A 3.2:** In activation steering, **the positive and negative examples are directly specified by the labels provided in each dataset. We extract activations from these labeled inputs accordingly.** For UltraFeedback and TruthfulQA, each training instance consists of a question paired with both positive and negative answers. We concatenate the question with each corresponding answer and feed the full sequence into the LLM to collect activations for the positive and negative cases. For the detoxification task, we directly use the provided toxic and non-toxic prompts as inputs for activation extraction. Further details are provided in lines 975–983 of the revised manuscript (page 19).
>
> **Q 3.3: Unverified barrier property: It is not shown that the learned barrier function (h(a)) satisfies Proposition ($\dot{h}(a) = \nabla h(a)^{\top}v(a) > 0$) in practice.**
>
> **A 3.3:** We show that the inequality $\dot{h}(a) = \nabla h(a)^{\top} v(a) > 0$ is ensured from both theoretical and empirical perspectives.
>
> **i) Theoretical perspective.** As defined in Eq. (13), $v(a)$ is set to $\frac{\nabla h(a)}{\| \nabla h(a) \|}$. Substituting this into the expression for $\dot{h}(a)$ yields $\dot{h}(a) = \nabla h(a)^{\top} \frac{\nabla h(a)}{\| \nabla h(a) \|}= \frac{\| \nabla h(a) \|^{2}}{\| \nabla h(a) \|} = \| \nabla h(a) \| \ge 0$, where the equality holds only when $\nabla h(a) = 0$. Since $h(a) = w^{\top} \phi(a) + b$ (Eq. 12), the condition $\nabla h(a) = 0$ requires either (i) $w = 0$ or (ii) the Jacobian $J_{\phi}(a) = 0$. In practice, both scenarios are highly unlikely: $w$ is obtained from logistic regression, and $\phi(a)$ is computed via polynomial count sketching, both of which produce non-zero values under normal training conditions. Hence, $\dot{h}(a) > 0$ holds almost everywhere in practice. We additionally provide a formal proof in Appendix C.4 for absolute clarity.
>
> **ii) Empirical perspective.** We additionally verify this property empirically by visualizing the evolution of $h(a)$ along BODES trajectories in the revised manuscript (see **Fig. 2 in Appendix C.4**, page 16). As shown in the figure, $h(a)$ consistently increases along the ODE trajectories, confirming $\dot{h}(a) > 0$ holds in practice.
>
> **Q 3.4: No trajectory-level likelihood analysis: The paper doesn’t measure or visualize how average likelihood ((h(a))) changes along the ODE trajectory.**
>
> **A 3.4:** We provide a trajectory-level visualization of $h(a)$ along the ODE trajectories in **Fig. 2 in Appendix C.4** for TruthfulQA using Llama3.1-8B. As shown in the figure, the value of $h(a)$ consistently increases along the ODE trajectory. This behavior is directly guaranteed by the construction of BODES, as explained in **A 3.3**.
>
>
> **Q 3.5: Could this framework generalize to alignment to reward-based alignment, comparison with self consistency**
>
> **A 3.5:** Our work focuses on developing a **unified framework for activation steering**, **as clearly stated in the title, abstract, and introduction of the paper**, . Reward-based alignment methods (e.g., RLHF) and self-consistency techniques **fall into entirely different methodological categories**. We will examine potential conceptual connections in future work, but such extensions are far outside the intended focus of this paper.

---

> ### Author Response · Authors · 2025-11-25
> **Request for Follow-Up Feedback on Author Rebuttal**
>
> Dear Reviewer BqZv,
>
> Our detailed rebuttal has been submitted, and we have thoroughly addressed all the points and suggestions you raised. We understand the significant workload involved in reviewing papers, but we kindly request your feedback on our responses to ensure that the discussions are as productive and comprehensive as possible. Your insights will be invaluable in refining the final version of our work. Thank you once again for your time and effort!

---

### Official Review · Reviewer_cs8e · 2025-11-02

**Soundness:** 3
**Presentation:** 3
**Contribution:** 3
**Rating:** 6
**Confidence:** 4

**Summary:**

This paper studies activation steering for large language models (LLMs) and reframes it as solving an ordinary differential equation (ODE). It shows that standard one-step activation addition can be viewed as the Euler step of this ODE, providing a unified perspective that connects both input reading and output optimization approaches via a barrier function over activations. The authors propose BODES, which learns a nonlinear log-density-ratio barrier between positive and negative activations using logistic regression on polynomial count-sketch features, and performs multi-step ODE integration along the barrier gradient for adaptive steering. Experiments on Falcon‑7B, Mistral‑7B, and Llama‑3.1‑8B show consistent improvements over one-step baselines, with average gains of about 7% on TruthfulQA and 2% on UltraFeedback and RealToxicityPrompts

**Strengths:**

1. The barrier-function perspective is well-grounded in control theory, offering a principled view that unifies existing steering methods.

2. The paper is clearly written, and empirical results are promising, showing consistent gains across multiple models and datasets.

**Weaknesses:**

1. The claimed advantage over output-optimization methods is debatable: both approaches rely on a learned scoring function—the proposed method’s barrier function also requires accurate estimation and introduces additional hyperparameters (e.g., step size, number of ODE steps, solver type, and polynomial sketch settings)


2. It is unclear whether the ODE formulation is necessary. Could similar results be achieved by taking several gradient ascent steps on the barrier function, which might be more efficient?

3. Inference efficiency is not discussed. Solving an ODE can be computationally expensive; a comparison of inference speed versus existing one-step or gradient-based steering methods would strengthen the paper.

4. How sensitive is performance to the ODE solver choice and step size? Does the performance degrade significantly if the ODE is approximated with fewer steps?

**Questions:**

See Weaknesses section.

---

> ### Author Response · Authors · 2025-11-21
> **Rebuttal by Authors**
>
> Dear Reviewer cs8e,
>
> We sincerely thank you for taking the time to review our paper and for providing insightful and constructive feedback. We have carefully addressed all the points and suggestions you raised. Please feel free to let us know if you have any additional questions or concerns. We greatly appreciate the opportunity to further refine our work. Thank you again for your thoughtful review and valuable input.
>
> **Q 2.1: The claimed advantage over output-optimization methods is debatable: both approaches rely on a learned scoring function—the proposed method’s barrier function also requires accurate estimation and introduces additional hyperparameters (e.g., step size, number of ODE steps, solver type, and polynomial sketch settings)**
>
> **A 2.1:** As mentioned in Section 4.2.2 and Appendices C.2–C.3, we clarify the advantages of BODES over output-optimization approaches by using RE-Control, the output-optimization baseline in our comparisons, as a representative example.
>
> 1. **Fundamentally different and more efficient learning of the scoring function:** Although both paradigms rely on a learned scoring function, the mechanisms differ substantially. **RE-Control learns its scoring function through an additional large pretrained reward model (13B parameters)**. Such reward models are typically trained for helpfulness and **may not generalize well to other alignment dimensions** such as truthfulness, which is reflected in Table 2 of our submission: RE-Control yields only marginal improvements on TruthfulQA. In contrast, BODES learns its barrier function directly from activation distributions via density ratio estimation. This can **be implemented easily using logistic regression**, eliminating the need for a large reward model and **allowing the method to capture intrinsic structure in activation distributions**. As a result, BODES achieves consistent improvements across all evaluated tasks.
> 2. **Higher computational efficiency:** As discussed in lines 282-285 and in **A 2.3**, output-optimization methods are generally less computationally efficient because **their scoring functions are based on DNNs.** In contrast, BODES employs a barrier function learned through **density ratio estimation using traditional machine learning techniques**, resulting in lower computational overhead. This observation is also supported by the results in **Tab 2.1**.
> 3. **Reduced hyperparameter tuning:** As described in Appendix C.2 and C.3., while BODES introduces several hyperparameters, **all of them except the intervention strength $T$ are kept the same across all datasets and LLMs**,  Even under this unified configuration, BODES consistently outperforms SOTA steering baselines. In contrast, RE-Control requires extensive dataset- and model-specific hyperparameter tuning to achieve comparable results.
>
> **Q 2.2: It is unclear whether the ODE formulation is necessary. Could similar results be achieved by taking several gradient ascent steps on the barrier function, which might be more efficient?**
>
> **A 2.2:** First, we would like to highlight that **the ODE formulation serves as the key theoretical contribution of this work**. It allows the incorporation of the barrier function from control theory, providing a unified and principled framework that connects and generalizes existing activation steering methods.
>
> Second, we note that **gradient ascent is mathematically equivalent to numerically solving an ODE using the Euler method**. Concretely, for $\max_{a} h(a)$, the gradient ascent update rule $a_{k+1} = a_k + s \cdot \nabla h(a_k)$ is equivalent to the Euler discretization of the ODE $\dot{a} = \nabla h(a)$. Therefore, gradient ascent can be regarded as a first-order approximation of the ODE formulation, and the two approaches **share comparable efficiency**. The connection between optimization algorithms and ODEs has also been well-established in prior literature [1, 2, 3].
>
> [1] Helmke et al. "Optimization and dynamical systems." Springer Science & Business Media, 2012.
>
> [2] Su et al. "A differential equation for modeling Nesterov's accelerated gradient method: Theory and insights." JMLR 2016.
>
> [3] Li et al. "The physical systems behind optimization algorithms." NeurIPS 2018.

---

> ### Author Response · Authors · 2025-11-21
> **Rebuttal by Authors (Continued)**
>
> **Q 2.3: Inference efficiency is not discussed. Solving an ODE can be computationally expensive; a comparison of inference speed versus existing one-step or gradient-based steering methods would strengthen the paper.**
>
> **A 2.3:** We compare the inference efficiency of BODES with other baseline methods on TruthfulQA. **The number of generated tokens per second** for each method is presented in **Tab 2.1**. The results show that the generation speed of BODES is **only slightly slower than the no-steering (original) condition**  and other one-step steering methods such as CAA, which is expected due to the multi-step nature of BODES.
>
> However, BODES is still **faster than several DNN-based steering methods**, including RE-Control and TruthFlow, which involve additional DNN computations during inference. These findings indicate that BODES maintains practical inference efficiency while introducing only minimal additional computational cost. We also include these results in Appendix E.2 (page 20, line 1069-1098) of the revised manuscript.
>
> **Tab 2.1 The number of generated tokens per second for three LLMs (higher is better).**
>
> | Model        | Falcon-7B     | Mistral-7B    | Llama3.1-8B   |
> |:-------------|:--------------|:--------------|:--------------|
> | Original     | 117.69 ± 0.45 | 116.26 ± 0.24 | 114.82 ± 0.18 |
> | RepE         | 117.69 ± 0.27 | 115.82 ± 0.08 | 114.71 ± 0.3  |
> | ITI          | 117.54 ± 0.12 | 115.78 ± 0.07 | 114.82 ± 0.29 |
> | CAA          | 117.46 ± 0.44 | 115.76 ± 0.03 | 114.57 ± 0.48 |
> | MiMiC        | 105.62 ± 0.36 | 109.66 ± 0.16 | 108.73 ± 0.38 |
> | HPR          | 116.09 ± 0.04 | 115.07 ± 0.12 | 114.42 ± 0.11 |
> | RE-Control   | 98.03 ± 0.51  | 101.05 ± 0.03 | 99.94 ± 0.11  |
> | LinAcT       | 117.61 ± 0.17 | 116.17 ± 0.03 | 115.0 ± 0.42  |
> | TruthFlow    | 62.45 ± 0.38  | 62.06 ± 0.46  | 62.33 ± 0.48  |
> | BODES (Ours) | 107.41 ± 0.22 | 105.89 ± 0.08 | 106.76 ± 0.06 |
>
>
> **Q 2.4: How sensitive is performance to the ODE solver choice and step size? Does the performance degrade significantly if the ODE is approximated with fewer steps?**
>
> **A 2.4:** We first assess the sensitivity of BODES to the choice of ODE solver on TruthfulQA, using True$\times$Info as the evaluation metric. The results, reported in **Tab. 2.2**, show that **BODES is largely insensitive to solver choice**. Higher-order solvers such as fourth-order Runge-Kutta method (RK4) yield only marginal improvement compared to simpler methods. Considering simplicity and computational efficiency, we therefore adopt the Euler method as our default solver.
>
> We next analyze the effect of the number of integration steps while keeping the intervention strength $T$ fixed. The results, presented in **Fig. 4 in Appendix E.4** (page 22), show that increasing the number of steps (i.e., reducing the step size) yields mild improvements initially, after which performance stabilizes. Importantly, **reducing the number of steps does not significantly degrade performance**, indicating that BODES remains robust even under coarse discretization. This robustness arises because the barrier function provides a consistent and reliable steering direction.
>
> Overall, these analyses demonstrate that BODES maintains stable performance across different ODE solvers and step sizes. These experimental results are included in the Appendix E.4 of the revised manuscript.
>
> **Tab 2.2: The impact of different ODE solver types on the True$\times$Info (%) performance of BODES on TruthfulQA.**
>
> | ODE Solver Type | Falcon-7B-Base   | Mistral-7B-Base   | Llama3.1-8B-Base   |
> |:------|:-----------------|:------------------|:-------------------|
> | Euler | 42.2 ± 0.115     | 59.9 ± 0.237      | 63.2 ± 0.823       |
> | RK4   | 42.8 ± 0.555     | 60.2 ± 1.328      | 63.3 ± 0.923       |

---

> ### Author Response · Authors · 2025-11-25
> **Request for Follow-Up Feedback on Author Rebuttal**
>
> Dear Reviewer cs8e,
>
> Our detailed rebuttal has been submitted, and we have thoroughly addressed all the points and suggestions you raised. We understand the significant workload involved in reviewing papers, but we kindly request your feedback on our responses to ensure that the discussions are as productive and comprehensive as possible. Your insights will be invaluable in refining the final version of our work. Thank you once again for your time and effort!

---

### Official Review · Reviewer_9xYB · 2025-11-05

**Soundness:** 2
**Presentation:** 3
**Contribution:** 3
**Rating:** 6
**Confidence:** 3

**Summary:**

This paper focuses on activation steering. It innovatively proposes a unified ordinary differential equation (ODE)–based theoretical framework for activation steering, and introduces multi-step activation steering derived from this formulation. The approach outperforms traditional one-step steering methods across multiple models and tasks.

**Strengths:**

1. The proposed ODE-based multi-step activation steering is well-motivated and demonstrates a meaningful degree of novelty.


2. The experiments cover multiple models and multiple tasks, clearly showing that the proposed method outperforms one-step activation steering baselines.

**Weaknesses:**

1. The proposed method requires multi-step ODE integration, which introduces additional computational overhead during inference. The paper would benefit from a more detailed analysis and discussion of this extra cost.


2. Some implementation details are insufficiently specified. During training, the method relies on collecting positive and negative activations, but the paper does not clearly describe how these activations are extracted, particularly which token positions are used. Additionally, during inference, steering is applied at every decoding step, it is unclear whether training is consistent with this procedure, i.e., whether activations used for training were also collected across different token positions or only from specific tokens.


3. It is unclear how well the proposed method transfers across datasets or domains. Additional experiments demonstrating whether the proposed method beyond the datasets it was trained on would help clarify the applicability of the approach.


4. Since the paper primarily focuses on alignment tasks, it is important to assess whether the method degrades model utility. Evaluating performance on general-purpose benchmarks such as MMLU would allow for a clearer understanding of whether the proposed steering affects core model capabilities.

**Questions:**

See Weaknesses

---

> ### Author Response · Authors · 2025-11-21
> **Rebuttal by Authors**
>
> Dear Reviewer 9xYB,
>
> We sincerely thank you for taking the time to review our paper and for providing insightful and constructive feedback. We have carefully addressed all the points and suggestions you raised. Please feel free to let us know if you have any additional questions or concerns. We greatly appreciate the opportunity to further refine our work. Thank you again for your thoughtful review and valuable input.
>
> **Q 1.1: The paper would benefit from a detailed analysis and discussion of additional computational overhead during inference.**
>
> **A 1.1:** We compare the inference efficiency of BODES with other baseline methods on TruthfulQA. **The number of generated tokens per second** for each method is presented in **Tab 1.1**. The results show that **the generation speed of BODES is only slightly slower than the no-steering (original) condition**  and other one-step steering methods such as CAA, which is expected due to the multi-step nature of BODES.
>
> However, **BODES is still faster than several DNN-based steering methods**, including RE-Control and TruthFlow, which involve additional DNN computations during inference. These findings indicate that BODES maintains practical inference efficiency while introducing only minimal additional computational cost. We also include these results in Appendix E.2 (page 20, line 1069-1098) of the revised manuscript.
>
> **Tab 1.1 The number of generated tokens per second for three LLMs (higher is better).**
>
> | Model        | Falcon-7B     | Mistral-7B    | Llama3.1-8B   |
> |:-------------|:--------------|:--------------|:--------------|
> | Original     | 117.69 ± 0.45 | 116.26 ± 0.24 | 114.82 ± 0.18 |
> | RepE         | 117.69 ± 0.27 | 115.82 ± 0.08 | 114.71 ± 0.3  |
> | ITI          | 117.54 ± 0.12 | 115.78 ± 0.07 | 114.82 ± 0.29 |
> | CAA          | 117.46 ± 0.44 | 115.76 ± 0.03 | 114.57 ± 0.48 |
> | MiMiC        | 105.62 ± 0.36 | 109.66 ± 0.16 | 108.73 ± 0.38 |
> | HPR          | 116.09 ± 0.04 | 115.07 ± 0.12 | 114.42 ± 0.11 |
> | RE-Control   | 98.03 ± 0.51  | 101.05 ± 0.03 | 99.94 ± 0.11  |
> | LinAcT       | 117.61 ± 0.17 | 116.17 ± 0.03 | 115.0 ± 0.42  |
> | TruthFlow    | 62.45 ± 0.38  | 62.06 ± 0.46  | 62.33 ± 0.48  |
> | BODES (Ours) | 107.41 ± 0.22 | 105.89 ± 0.08 | 106.76 ± 0.06 |
>
>
> **Q 1.2: The paper does not clearly describe how these activations are extracted, particularly which token positions are used.**
>
> **A 1.2:** **(i) Activation extraction procedure.** For UltraFeedback and TruthfulQA, each training instance consists of a question paired with both positive and negative answers. We concatenate the question with each corresponding answer and feed the full sequence into the LLM to collect activations for the positive and negative cases. For the detoxification task, the Jigsaw dataset does not contain explicit questions; therefore, we directly use the provided toxic and non-toxic prompts as inputs for activation extraction.
>
> **(ii) Token positions used during training.** Consistent with common practice in activation steering [1, 2, 3, 4], we extract activations from the **last token position** of each input sequence at the residual stream of a specific layer (layer selection is discussed in Appendix D.2). This design matches the inference-time procedure, where steering is always applied at every decoding step to the newly generated token’s activation. We have added clarifications regarding this procedure in the revised manuscript (see page 19, line 975-983).
>
> [1] Wehner et al. "Taxonomy, opportunities, and challenges of representation engineering for large language models." arXiv:2502.19649.
>
> [2] Rimsky et al. "Steering Llama 2 via Contrastive Activation Addition." ACL 2024.
>
> [3] Li et al. "Inference-time intervention: Eliciting truthful answers from a language model." NeurIPS 2023.
>
> [4] Singh et al. "Representation surgery: Theory and practice of affine steering." ICML 2024.

---

> ### Author Response · Authors · 2025-11-21
> **Rebuttal by Authors (Continued)**
>
> **Q 1.3: It is unclear how well the proposed method transfers across datasets or domains.**
>
> **A 1.3:** To assess the transferability of BODES across datasets and domains, we trained BODES on TruthfulQA using Llama3.1-8B and then directly applied it (without any additional tuning) to the multiple-choice benchmark CommonsenseQA [5] in a **zero-shot** manner. The results, shown in **Tab. 1.2**, indicate that **BODES achieves a slight performance improvement on CommonsenseQA**. We attribute this to the fact that CommonsenseQA also reflects aspects of truthfulness, though it focuses more on everyday common knowledge. These findings suggest that BODES can generalize beyond its training domain.
>
> **Tab 1.2: Accuracy of Llama3.1-8B with and without BODES on CommonsenseQA**
> |                     | CommonsenseQA |
> |---------------------|---------------|
> | Llama3.1-8B         | 68.0%         |
> | Llama3.1-8B + BODES | **68.3%**     |
>
> [5] CommonsenseQA: https://huggingface.co/datasets/tau/commonsense_qa
>
>
> **Q 1.4: Since the paper primarily focuses on alignment tasks, it is important to assess whether the method degrades model utility.**
>
> **A 1.4:** Following the evaluation setup described in A 1.3, we assess whether BODES affects the general capabilities of LLMs by testing performance on MMLU [6]. As shown in Tab. 1.3, **BODES does not introduce significant degradation on this benchmark.** We note that some degree of degradation is common in activation-steering approaches. For example, Lin-Act [7], one of our comparison baselines, also leads to a reduction in utility. Nonetheless, BODES maintains competitive performance on MMLU, indicating that its steering behavior does not substantially harm the model’s core general-purpose abilities. Experimental results in A 1.3 and A 1.4 are included in Appendix E.3 of the revised manuscript.
>
> **Tab 1.3: Accuracy of Llama3.1-8B with and without BODES on MMLU.**
> |                     | MMLU  |
> |---------------------|-------|
> | Llama3.1-8B         | 61.8% |
> | Llama3.1-8B + BODES | 60.9% |
>
> [6] MMLU: https://huggingface.co/datasets/cais/mmlu
>
> [7] Rodriguez et al. "Controlling language and diffusion models by transporting activations." ICLR 2025.

---

> ### Author Response · Authors · 2025-11-25
> **Request for Follow-Up Feedback on Author Rebuttal**
>
> Dear Reviewer 9xYB,
>
> Our detailed rebuttal has been submitted, and we have thoroughly addressed all the points and suggestions you raised. We understand the significant workload involved in reviewing papers, but we kindly request your feedback on our responses to ensure that the discussions are as productive and comprehensive as possible. Your insights will be invaluable in refining the final version of our work. Thank you once again for your time and effort!

---

### Author Response · Authors · 2025-11-21
**General Response to All Reviewers**

Dear Reviewers,

We sincerely thank you for taking the time to review our paper and for providing insightful and constructive comments. We have carefully addressed all the points and suggestions you raised and have **submitted a revised manuscript, with all substantial changes highlighted in blue for ease of reference**. Please do not hesitate to let us know if you have any additional questions or concerns. We greatly appreciate the opportunity to further refine our work and would be grateful for any further feedback you may have. Thank you once again for your thoughtful review.

---

### Meta-Review · Area_Chair_bPwR · 2026-01-07

**Summary:**

Reviewers highlighted the paper’s novel ODE-based framework, which provides a principled perspective on activation steering methods, and acknowledged its consistent performance across multiple LLMs and datasets. They raised concerns about inference efficiency, sensitivity to ODE solver choices, step size, intervention strength, transferability, and potential impact on LLM utility. In response, the authors conducted additional experiments that addressed these points, demonstrating that the method is efficient, robust to hyperparameter choices, and maintains strong performance and transferability without negatively affecting LLM utility.

**Reviewer Concerns:**

The rebuttal has effectively addressed concerns regarding the inference efficiency of the proposed method, its sensitivity to ODE solver choices, step size, and intervention strength, as well as its transferability and potential impact on LLM utility. In my view, there are no significant outstanding issues.

**Reviewer Scores:**

Three reviewers initially gave the paper a positive rating of 6, and it is likely they will maintain or even improve their ratings after considering the author’s response.

The only reviewer who gave a negative initial rating of 2 is also likely to improve their rating, as the author’s response appears convincing and reasonable to me.

---

### Decision · Program_Chairs · 2026-01-26

Accept (Poster)